# Structural Basis of the Avian Influenza NS1 Protein Interactions with the Cell Polarity Regulator Scribble

**DOI:** 10.3390/v14030583

**Published:** 2022-03-11

**Authors:** Airah Javorsky, Patrick O. Humbert, Marc Kvansakul

**Affiliations:** 1Department of Biochemistry and Chemistry, La Trobe Institute for Molecular Science, La Trobe University, Melbourne, VIC 3086, Australia; 18583387@students.latrobe.edu.au; 2Research Centre for Molecular Cancer Prevention, La Trobe University, Melbourne, VIC 3086, Australia; 3Department of Biochemistry and Pharmacology, University of Melbourne, Melbourne, VIC 3010, Australia; 4Department of Clinical Pathology, University of Melbourne, Melbourne, VIC 3010, Australia

**Keywords:** Influenza A virus, bird-flu, H5N1, cell polarity, X-ray crystallography, isothermal titration calorimetry, NS1, PDZ, scribble

## Abstract

Scribble is a highly conserved regulator of cell polarity, a process that enables the generation of asymmetry at the cellular and tissue level in higher organisms. Scribble acts in concert with Disc-large (Dlg) and Lethal-2-giant larvae (Lgl) to form the Scribble polarity complex, and its functional dysregulation is associated with poor prognosis during viral infections. Viruses have been shown to interfere with Scribble by targeting Scribble PDZ domains to subvert the network of interactions that enable normal control of cell polarity via Scribble, as well as the localisation of the Scribble module within the cell. The influenza A virus NS1 protein was shown to bind to human Scribble (SCRIB) via its C-terminal PDZ binding motif (PBM). It was reported that the PBM sequence ESEV is a virulence determinant for influenza A virus H5N1 whilst other sequences, such as ESKV, KSEV and RSKV, demonstrated no affinity towards Scribble. We now show, using isothermal titration calorimetry (ITC), that ESKV and KSEV bind to SCRIB PDZ domains and that ESEV unexpectedly displayed an affinity towards all four PDZs and not just a selected few. We then define the structural basis for the interactions of SCRIB PDZ1 domain with ESEV and ESKV PBM motifs, as well as SCRIB PDZ3 with the ESKV PBM motif. These findings will serve as a platform for understanding the role of Scribble PDZ domains and their interactions with different NS1 PBMs and the mechanisms that mediate cell polarity within the context of the pathogenesis of influenza A virus.

## 1. Introduction

Cell polarity refers to the differential distribution of macromolecules within a cell that allows it to orientate itself in a specific direction, which is an essential property for correct tissue development, organisation and function [1]. Examples of cell polarity include the axonal-dendritic directionality of neurons, the migration and asymmetric division of mesenchymal cells, and the apical-basal orientation or planar polarity of epithelial cells [2,3,4,5,6]. These phenomena can be categorised into the four main types of cell polarity as follows: (1) asymmetric cell division (ACD), (2) planar cell polarity (PCP), (3) apical-basal cell polarity (ABCP) and (4) front–rear cell polarity (FRCP) [2,3,4,5,6]. The proper establishment of cell polarity is crucial for normal cell homeostasis. With the loss of cell polarity, tissue becomes disorganised and excessive cell proliferation can present itself as a hallmark of cancer or viral replication [5,7]. For apical-basal, migrating and asymmetrical dividing polarity cell types three key protein complexes are involved in an antagonistic relationship to control cell polarity: the PAR module, the Crumbs module and the Scribble module [8]. These polarity regulatory modules play important roles in determining cell architecture and are highly conserved in higher organisms.

The Scribble module comprises three scaffolding proteins known as Scribble (Scrib), Discs-large (Dlg) and lethal-2-giant larvae (Lgl) [9,10,11]. These proteins are localised in the cytoplasm, near the plasma membrane, and are distributed asymmetrically within the cell. In epithelial cells, the PAR and Crumbs modules are localised along the apical membrane, whereas the Scribble module is found at the lateral membrane [11]. Scrib, Dlg and Lgl contain several protein interaction domains that allow them to bind to important cellular interactors, such as kinases and phosphatases, as a means of regulating cell signalling; however, their function is also dependent on their cellular localisation [11]. Scrib belongs to the LAP (leucine-rich repeats and PDZ domain) protein family, and features sixteen LLRs (Leucine-rich repeats), two LAP-specific domains (LAPSDa and LAPSDb) and the four PDZ domains (PDZ1, PDZ2, PDZ3 and PDZ4) [12]. In particular, the PDZ domains have been identified to be the key mediators for the vast majority of Scribble and ligand interactions [11]. Scribble is localised with Dlg and Lgl in the epithelial tight junction at the apical and basolateral cell membrane, and has been shown to interact with several viral proteins which disturb the formation of adherens junctions and the interaction with Lgl [13]. Some of these viral proteins include the human papillomavirus (HPV) E6, human T-lymphotropic virus 1 (HTLV-1) Tax, tick-borne encephalitis virus (TBEV) non-structural 5 (NS5) and influenza A virus NS1, with the latter interacting with the PDZ domains of Scribble [13,14,15,16,17].

PDZ domains are usually approximately 90 amino acids in size and consist of 5–6 β-strands and 2–3 α-helices with a conserved fold comprising a canonical ligand-binding groove formed by the α2 helix and the β2, β4 and β5 strands. This binding groove allows PDZ domains to recognise short amino acid motifs at the extreme C-termini of ligand proteins, coined PDZ binding motifs (PBMs); however, certain PDZ domains have also been shown to recognise internal sequence motifs [18,19]. Despite this interaction conservation, Scribble demonstrates a high degree of selectivity in target ligand interaction profiles [11]. PDZ domains are divided into three major classes based on their ligand recognition sequences, as follows: Class I has an X-T/S-X-V/L_COOH_ motif, Class II with an X-ψ-X-ψ_COOH_ motif, and Class III interacting with X-D/E-X-ψ_COOH_ motif [20], in which X is an unspecified amino acid and ψ refers to any hydrophobic amino acid. Scribble PDZ domains belong to the Class I PDZ domains and have been shown to interact with micromolar affinities, display overlapping preferences towards similar ligands and interact with numerous binding partners [21]. The specificity of these interactions is based on the PBM sequence, and the numbering is denoted as position 0 being the last C-terminus residue followed by the numbering of the consecutive −1 amino acids moving away from the ground 0 position, for example, “…–(−3)–(−2)–(−1)–(0)COOH” [21].

Non-transforming viruses, such as the tick-borne encephalitis virus (TBEV) and influenza A virus, have been found to encode PBMs that target a range of PDZ domains to aid dissemination in the host and transmission to new hosts as a means of enhancing viral replication [13]. For influenza A virus, interactions of the non-structural protein 1 (NS1) with Scribble have been reported [17,22]. Influenza A virus NS1 is a 26 kDa protein involved in multiple mechanisms to aid viral replication, such as preventing polyadenylation and splicing of cellular mRNA [23]. NS1 contains an N-terminal RNA binding domain that counters activation of the interferon response, which halts cellular signalling in the event of detection of a pathogenic presence, and a C-terminal effector domain that inhibits CPSF30 (cleavage and polyadenylation specificity factor), which is important for processing and export of mRNA [23]. The human influenza A virus protein NS1 has been shown to harbour multiple C-terminal sequences that conform to the following consensus sequence of class I PDZ binding motifs: RSKV (seen in H3N2), ESKV (found in H9N2, H5N1, H5N2), KSEV (associated with H1N1), ESEV (from H5N1) and RSEV (from H1N1) [22,23]. These sequences all appear to contribute to the pathogenicity of their corresponding strains; however, their impact on pathogenicity and virulence varies, potentially due to differences in the interactions and affinities of the PBM sequences. Whilst the four C-terminal residues of NS1 were shown to make species-specific contributions to influenza A virus virulence [24], sequence variation has been seen at the NS1 C-terminus, including premature termination, which yields NS1 proteins that lack PBM sequences. For avian influenza A NS1, ~90% of isolates contain an ESKV or ESEV PBM motif at the C-terminus [25], whereas in human influenza A the RSKV PBM motif is more widely found [26]. Studies using pull-down assays have shown that ESEV PBM of the avian influenza A virus H5N1 NS1 protein binds to the PDZ domains of Scribble, Dlg1 and various other PDZ domain proteins (MAGI-1, MAGI-2 and MAGI-3); however, it has not been clarified which specific PDZ domain(s) are involved in the interaction [17]. The disruption of Scribble, and other PDZ domains containing polarity proteins, also disables tight junctions (TJ) formation and supports viral replication, by enhancing dissemination to host cells, and viral spread [17]. The severity of the H5N1 influenza A virus can range from diarrhoea, encephalitis, cytokine storm, oedema within the lungs to organ failure and death [27]. Understanding the mechanism that underlies the avian influenza A virus H5N1 NS1 interactions and disruption of PDZ domain mediated signalling will enable a better understanding of the pathogenesis of H5N1.

## 2. Materials and Methods

### 2.1. Protein Expression and Purification

The synthetic codon optimised cDNAs encoding for the human Scribble (SCRIB) PDZ1 domain (residues 725–815) and SCRIB PDZ2 domain (residues 860–950) were cloned into the pGex-6P3 vector (Bioneer, Daejeon, Korea) that contained a Glutathione S-transferase (GST)-tag encoding sequence at the N-terminus followed by a human rhinovirus (HRV) 3C protease cleavable linker, as described previously [28]. cDNAs of SCRIB PDZ3 (residues 1002–1093) and SCRIB PDZ4 (residues 1099–1203) were cloned into the pGil vector (Bioneer) which contained a Hexahistidine-Maltose binding protein (His6-MBP) tag at the N-terminus followed by a tobacco etch virus (TEV) protease cleavable linker described previously [28]. SCRIB PDZ mutants cDNA PDZ3 H1071A (residues 1002–1093), PDZ4 R1110D (residues 1099–1203), PDZ4 R1116D (residues 1099–1203) and PDZ4 H1170A (residues 1099–1203) were cloned into the pGex-6P3 vector (Bioneer). SCRIB PDZ3 K1040A (residues 1002–1093) was described previously [29], and all other PDZ mutant constructs were also described previously [30].

The transformed E. coli BL21 (DE3) pLysS cells were grown in 2YT media (1.6% *w*/*v* Tryptone, 1.0% *w*/*v* Yeast Extract and 0.5% *w*/*v* sodium chloride (NaCl)), supplemented with 100 μg/mL ampicillin; cells were incubated at 37 °C and shaken at 160 rpm for approximately 20 h. Expression of all PDZ constructs was induced using an autoinduction protein expression protocol by growing the small-scale bacterial culture in 2YT media with 100 mM NaCl, 10 mM Tris pH 7.6, 1 mM MgSO_4_, 5052 solution (25% glycerol, 10% lactose and 2.5% glucose) and 100 µg/mL ampicillin, which was used for a large-scale cell culture by incubated shaking at 37 °C 160 rpm for approximately 4 h and then at 16 °C for at least 20 h [31]. For His-MBP-tagged affinity purification, proteins were purified via immobilised metal affinity chromatography (IMAC), using HisTrap HP columns (GE Healthcare, Chicago, IL, USA). GST-tagged proteins were purified using glutathione Sepharose 4B resins (GE Healthcare) equilibrated with GST purification buffer. All proteins were purified as previously described [28,29,30].

### 2.2. Circular Dichroism (CD) Spectroscopy

All proteins were diluted to 0.150 mg/mL in phosphate buffer (50 mM phosphate) and subjected to CD spectroscopy using the AVIV model 420 CD spectrometer at a wavelength from 190 to 260 nm with sample entry every 1 nm and an averaging time of 4 s at 25 °C. Data were processed using the AVIV Biomedical software and plotted using Microsoft Excel.

### 2.3. Isothermal Titration Calorimetry (ITC)

Binding of the NS1 peptides (Mimotopes Pty Ltd., Melbourne, Australia) to SCRIB PDZ constructs was determined by ITC with a range of influenza A virus strain PBM peptide sequences (Table 1); KMARTIESEV was derived from H5N1 A/Viet Nam/1194/2004 virus (Uniprot accession code Q6DP93, residues 216–225) (Mimotopes), KMARTARSKV was from H3N2 A/Memphis/14/1998 strain (Uniprot accession code Q2LN47, residues 221–230) (Mimotopes), KMARTIKSEV was from the H1N1 A/Brevig Mission/1/1918 strain (Uniprot accession code Q99AU3, residues 221–230) (Mimotopes), KMARTIESKV PBM is from the H5N1 A/chicken/Hubei/327/2004 strain, which was similar to the A/duck/Hunan/69/2004 strain (Uniprot accession code Q6B3P2, residues 216–225) (Mimotopes) [17,32]. The EMAGTIRSEV peptide was from the A/Puerto Rico/8/1943/H1N1 strain (Uniprot accession code P03496, residues 221–230) (Mimotopes). As a negative control, the KMERTIEPEV sequence, which is not a PBM sequence and was obtained from the H5N1 A/Hong Kong/156/97 strain (Uniprot accession code O56264, residues 221–230) (Mimotopes), was used [17]. The NS1 peptide sequences are distinguished based on the last four C-terminal residues of their PBM. The superpeptide was originally identified via the screening of a phage-displayed peptide library and was shown to harbour affinity towards a wide range of different PDZ domains [33] including those from polarity regulatory proteins [28,34]. These specific sequences, similarities and chemical composition of the peptides used throughout the study can be seen in Figure 1.

ITC was conducted using a MicroCal iTC200 System (GE Healthcare) at 25 °C as previously described [28]. Peptides and PDZ domain proteins were diluted in their respective buffer (either Tris Buffer or HEPES Buffer) to concentrations of 900 µM and ~75 µM respectively. Data were processed with the Origin 7.0 software (OriginLab Corporation, Northampton, MA, USA) using a “one-site binding model” to compute the binding affinities and stoichiometries.

### 2.4. Crystallisation and Structure Determination

Complexes of SCRIB PDZ domains (Table 2) were reconstituted as described (see also [35]). Briefly, proteins and 8-mer peptides were mixed at a 1:4 molar ratio and then concentrated using 3 kDa cut-off Ultra-0.5 centrifugal filter units (Millipore, Burlington, MA, USA). Concentrated protein complexes were subjected to high-throughput crystallisation screening using a Gryphon LCP (Art Robbins Instruments, Sunnyvale, CA, USA) in-house at 20 °C and prepared as drops of 0.2 μL protein and 0.2 μL reservoir solution in 96-well sitting drop crystallisation plates (Swissci, High Wycombe, UK) with commercial sparse matrix screens; JCSG-plus: HT-96 sparse matrix screen (Molecular Dimensions, Altamonte Springs, FL, USA), ShotGun sparse matrix screen (Molecular Dimensions), PACT Premier HT-96 sparse matrix screen (Molecular Dimensions), Structure Screen 1 + 2 sparse matrix screen (Molecular Dimensions) and Salt Rx sparse matrix screen (Hampton Research, Aliso Viejo, CA, USA). Crystals were mounted on nylon and copper loops (MiTGen, Ithaca, NY, USA). All data were collected at the Australian Synchrotron using the MX2 beamline equipped with the Eiger 16M detector (Dectris, Baden, Switzerland) with an oscillation range 0.1° per frame with a wavelength of 0.9537, integrated using XDS [36] and scaled using AIMLESS [37].

SCRIB PDZ1:ESEV crystals were grown in 0.2M Ammonium sulfate, 30% *w*/*v* PEG 4000 and flash cooled at −173 °C neat. The SCRIB PDZ1:ESEV complex belongs to space group C2 with a = 56.88 Å, b = 57.01 Å, c = 61.62 Å, α = 90.00°, β = 117.39°, γ = 90.00°. Molecular replacement was carried out using PHASER [38] with the previously solved structure of SCRIB PDZ1:b-PIX (PDB ID: 5VWK) [28] as a search model. SCRIB PDZ1:ESEV crystals contained 2 molecules each of SCRIB PDZ1 and ESEV peptide in the asymmetric unit, with 26.81% solvent content. The final TFZ and LLG values after molecular replacement were 22.7 and 919.2 respectively. The final model was built manually over several cycles using Coot [39] and refined using PHENIX with a final R_work_/R_free_ of 0.25/0.28, with 96.83% of residues in the favoured region of the Ramachandran plot and 0% of rotamer outliers.

SCRIB PDZ1:ESKV crystals were grown in 2.5M Ammonium nitrate, 0.1M Sodium acetate trihydrate pH 4.6 and were flash cooled at −173 °C. The SCRIB PDZ1:ESKV complex crystals formed plate-shaped crystals that belong to space group C2 with a = 58.54 Å, b = 51.20 Å, c = 27.87 Å, α = 90.00°, β = 90.55°, γ = 90.00°. Molecular replacement was carried out using PHASER [38] with the previously solved structure of SCRIB PDZ1: b-PIX (PDB ID: 5VWK) [28] as a search model. SCRIB PDZ1:ESKV crystals contained 1 molecule of SCRIB PDZ1 and 1 molecule of ESKV peptide in the asymmetric unit, with 18.13% solvent content and final TFZ and LLG values of 12.8 and 374.7, respectively. Model building and refinement were performed as above with final R_work_/R_free_ of 0.27/0.29, with 95.70% of residues in the favoured region of the Ramachandran plot and no rotamer outliers.

SCRIB PDZ3:ESKV crystals were grown in 0.1M Sodium Cacodylate pH4.2, 40% MPD and were flash cooled at −173 °C with 20% ethylene glycol. The SCRIB PDZ3:ESKV complex crystals formed prism-shaped crystals and belong to space group C2 with a = 77.84 Å, b = 77.72 Å, c = 64.75 Å, α = 90.00°, β = 94.16 °, γ = 90.00°. Molecular replacement was carried out using PHASER [38] with the previously solved structure of SCRIB PDZ1: b-PIX (PDB ID: 5VWK) [28] as a search model. SCRIB PDZ1:ESKV crystals contained 4 molecules of SCRIB PDZ1 and 4 molecules of ESKV peptide in the asymmetric unit, with 45.75% solvent content and final TFZ and LLG values of 17.2 and 411.03, respectively. Model building and refinement were performed as above with final R_work_/R_free_ of 0.257/0.303, with 99.08% of residues in the favoured region of the Ramachandran plot and no rotamer outliers.

## 3. Results

### 3.1. Biochemical Characterisation of SCRIB PDZ Domains and NS1 Peptide Interactions

NS1 interactions with SCRIB PDZ domains have been previously described in the literature [17,22,32] using immunofluorescence analysis, co-immunoprecipitation and GST pull-down assays. Although these experiments provided information regarding the relative binding of some of the PBM sequences to Scribble PDZ domains, no affinity measurements had been performed. To investigate these interactions and define their affinities, SCRIB PDZ domains were systematically evaluated using isothermal titration calorimetry (ITC) for binding to a range of NS1 PBM sequences (Table 1). As expected the EPEV peptide did not bind any of the SCRIB PDZ domains since it lacks a canonical PBM, similar to the RSKV peptide that also showed no binding, in agreement with previously published results [22]. Our ITC analysis showed the ESEV PBM from the highly pathogenic strain peptide is able to interact with all four individual SCRIB PDZ domains with affinities ranging from 11.8 ± 5.5 to 21.4 ± 1.7 µM (Figure 2, Table 2). Interestingly, these interactions contrast with previously observed interactions from GST pull-down studies that suggested the ESEV PBM sequence in the context of recombinant full-length NS1 was only able to bind to a tandem of PDZ1 and PDZ2 [22]. Examination of SCRIB PDZ domain interactions with the PBM peptide from the ESKV PBM revealed with SCRIB PDZ1, 2 and 3 domains, but not with PDZ4 (Table 2). KSEV, RSKV and RSEV showed no binding to the individual PDZ domains (Table 2). The negative control EPEV also showed no binding as expected (Table 2) [17].

### 3.2. Structural Analysis of SCRIB PDZ-NS1 Interactions

To gain structural insight into the interactions between the SCRIB PDZ domains and NS1 PBM motifs, we systematically attempted to crystallise complexes of individual SCRIB PDZ domains bound to identified interacting NS1 PBM peptides from our ITC analysis.

We determined crystal structures of SCRIB PDZ1 bound to NS1 ESEV PBM (A^281^-R^219^-T^220^-I^221^-E^222^-S^223^-E^224^-V^225^; Figure 3), SCRIB PDZ1 bound to NS1 ESKV PBM (A^281^-R^219^-T^220^-I^221^-E^222^-S^223^-K^224^-V^225^; Figure 3) and PDZ3 bound to NS1 ESKV PBM (A^281^-R^219^-T^220^-I^221^-E^222^-S^223^-K^224^-V^225^; Figure 3, Table 3). The Scribble PDZ domains adopt a compact globular fold comprising five to six β-strands and two α-helices that form a β-sandwich structure when engaged with the C-terminal PBM [28]. Examination of the SCRIB PDZ:NS1 complexes showed that the NS1 peptides are bound in the canonical ligand-binding groove formed by the β2 strand and helix α2 (Figure 3). A superimposition with the previously solved SCRIB PDZ1:β-PIX structure (PDB ID: 5VWK) on SCRIB PDZ1:ESEV yielded an overall root-mean-square deviation (RMSD) of 0.73 Å (Figure 4). SCRIB PDZ1:ESKV superimposed with SCRIB PDZ3:β-PIX (PDB ID: 5VW1) yielded an RMSD of 0.75 Å, and SCRIB PDZ1:ESKV superimposed with SCRIB PDZ3:β-PIX (light grey, PDB ID 5VW1) 0.57 Å.

The SCRIB PDZ1:ESEV complex revealed hydrogen bonds between Ser223^ESEV^-Ile742^SCRIB PDZ1^, Glu222^ESEV^-Ser762^SCRIB PDZ1^ and Ile223^ESEV^-Gly744^SCRIB PDZ1^ (Figure 3). The SCRIB PDZ1:ESKV complex revealed hydrogen bonds between Ser223^ESKV^-Ile742^SCRIB PDZ1^ and Ser223^ESKV^-His793^SCRIB PDZ1^ (Figure 3). In the SCRIB PDZ3:ESKV complex, Val225^ESKV^ is docked in the conserved hydrophobic pocket formed by SCRIB PDZ3 Leu1014, Gly1014 and Leu1016 (Figure 3). Additionally, hydrogen bonds are formed between Lys224^ESKV^-Leu1016^SCRIB PDZ3^, Ser223^ESKV^-Ile1018^SCRIB PDZ3^ and Ser223^ESKV^-His1071^SCRIB PDZ3^ at the α2 helix (Figure 3). Glu222^ESKV^ side-chain reaches over to the β2 strand and forms an ionic bond with the Lys1039^SCRIB PDZ2^.

To validate our crystal structures, we performed mutagenesis to probe SCRIB PDZ domain interactions with NS1 PBM in ITC. The PDZ1 mutant H793A abolished binding with both the ESEV and ESKV PBM (Table 1, Figure 5). SCRIB PDZ2 H928A bound to ESEV with a 5-fold tighter affinity (3.6 ± 0.7 µM) compared to SCRIB PDZ2 WT (18.7 ± 2.4 µM), and, interestingly, did not bind to ESKV (Table 1, Figure 5). SCRIB PDZ3 H1071A did not bind to either ESEV or ESKV, whereas PDZ3 K1040A bound to both (ESEV: 22.2 ± 1.1 µM) (ESKV: 13.8 ± 0.8 µM) (Table 1, Figure 5). None of our SCRIB PDZ4 mutants (PDZ4 R1110D, PDZ4 R1116D and PDZ4 H1170A) interacted with NS1 ESEV. We confirmed correct folding of all mutants using CD spectroscopy (Figure 6).

## 4. Discussion

To understand the ability of the influenza A virus protein NS1 to influence SCRIB signalling as an exogenous interaction partner of SCRIB, we systematically examined the interactions of individual SCRIB PDZ domains with the PBM peptides of several influenza A virus strains in vitro using ITC. The ESEV sequence showed an affinity towards all four SCRIB PDZ domains, with SCRIB PDZ4 showing the highest affinity interaction (K_D_ = 11.8 ± 5.5 µM), followed by SCRIB PDZ3 (K_D_ = 13.1 ± 1.6 µM), SCRIB PDZ2 (K_D_ = 18.7 ± 2.4 µM) and lastly SCRIB PDZ1 (K_D_ = 21.4 ± 1.7 µM). Our measurements were in contrast to previously reported GST pull-down assays [22], which suggested that NS1 ESEV was unable to bind to any of the single SCRIB PDZ domains and only interacted with SCRIB PDZ domains arranged as PDZ1 and PDZ2 tandem. However, we note that our ITC experiments were performed using 10-mer PBM peptides, whereas the GST pull-down study used full-length NS1 proteins, which may affect the accessibility of the PBM to the PDZ constructs [22].

The NS1 PBM sequence ESKV from the Hubei strain was originally identified via a phylogenetic analysis that identified a mutation in the NS1 protein from the H5N1 virus; the avian virus-type PBM sequence ESEV (which affects virulence) was replaced with ESKV [32]. Whilst NS1, featuring the ESKV sequence, supported respiratory infection in mice, the substitution of Lys to Glu resulted in a slight decrease in virulence [32]. Our ITC analysis of the ESKV PBM revealed an interaction with SCRIB PDZ2 (K_D_ = 9.5 ± 0.7 µM) as the tightest interactor, followed by SCRIB PDZ3 (K_D_ = 12.1 ± 0.9 µM) and SCRIB PDZ1 (K_D_ = 30.0 ± 0.9 µM) as the weakest interactor. Furthermore, our data also differ from other biochemical studies reported, where co-immunoprecipitation experiments showed only negligible binding to SCRIB [32]. Interestingly, our ITC measurements showed that a single amino acid difference in the PBM sequence (Glu acid to Lys) in position −1 abolishes the interaction with SCRIB PDZ4. To date there is no available structure of the individual SCRIB PDZ4 domain bound to an interactor; however, several studies have previously reported interactions of SCRIB PDZ4 with other ligands [14,40]. Notably, glutamic acid is negatively charged and the change in charge at position −1 raises the possibility that it is interacting with a positively charged residue within the SCRIB PDZ4 domain. With this possibility in mind, Arg1116 PDZ4 was a target for mutagenesis, since β2-strand is a typical location for the −1 position of a PBM to interact. By mutating its basic residue Arg1116 into an acidic residue like Asp, this mutation was shown to ablate the interaction of the ESEV peptide (Figure 5). This suggests the possibility that specific ionic interactions play an important role in NS1 binding to SCRIB PDZ4.

The majority of the SCRIB PDZ mutants displayed abolished binding (PDZ1 H793A, PDZ3 H1071A, PDZ4 R1110D, PDZ4 R1116D, PDZ4 H1170A) with ESEV and ESKV (Table 1, Figure 5). It appears that the conserved His in each domain found on the α2 helix, PDZ1 H793, PDZ3 H1071 and PDZ4 H1170A is an important position even when this was not obvious from the SCRIB PDZ1:ESEV structure, whereas in the SCRIB PDZ1:ESKV and SCRIB PDZ3:ESKV complexes a clear role for this His could be identified (Figure 3). In the case of the SCRIB PDZ1:ESEV complex, this is possibly due to the crystallographic packing (resulting in a different rotamer for the Ser 223 position in ESEV compared to ESKV), which prevents the Ser hydroxyl group from reaching SCRIB PDZ1 H793 for a hydrogen bond in the SCRIB PDZ1:ESEV complex (Figure 3). However, this interaction may be available in solution, since substitution of the His by an Ala significantly impacts the affinity of the interaction. Notably, SCRIB PDZ2 H928A bound to ESEV with a 5-fold tighter affinity (3.6 ± 0.7 µM) compared to PDZ2 WT (18.7 ± 2.4 µM), and did not bind to ESKV (Table 1, Figure 5). In the absence of a structure for a SCRIB PDZ2:NS1 PBM complex, we have no structural insight into how this amino acid change in SCRIB PDZ2 leads to such a difference in interactions between the ESEV and ESKV peptides, and why SCRIB PDZ2 H928A has an increased affinity for ESEV.

However, in previously solved structures, like SCRIB PDZ2:Vangl2 [30], this PDZ2 H928 is an important position for an interaction, which could explain the abolished binding seen with the ESKV peptide. SCRIB PDZ3 K1040A showed a weaker interaction with ESEV (22.2 ± 1.1 µM) and ESKV (13.8 ± 0.8 µM) compared to the wild-type SCRIB PDZ3 counterpart (ESEV:13.1 ± 1.6 µM, ESKV:12.1 ± 0.9 µM) (Table 1, Figure 5). Although SCRIB PDZ3 K1040 formed an ionic bond with ESKV seen in the SCRIB PDZ3:ESKV complex, it appears not to be a critical contributor to the overall affinity, with loss of the interaction only modestly reducing the overall affinity of the SCRIB PDZ3:PBM interaction. SCRIB PDZ3 H1071A revealed complete loss of binding for both ESEV and ESKV, whereas PDZ3 K1040A interacted with both (ESEV: 22.2 ± 1.1 µM) (ESKV: 13.8 ± 0.8 µM) (Table 1, Figure 5).

Considering that H5N1 originated in birds, we examined the protein sequence conservation of interacting human and avian Scribble PDZ domains (Figure 7). Examination of the main NS1 interaction, mediating secondary structure elements from Scrib PDZ domains, reveals that all residues involved in hydrogen bonds are conserved, as well as all main residues involved in van der Waals contacts. This suggests that the avian influenza A virus NS1 may be as effective at manipulating Scribble-mediated polarity signalling in avian species as in humans.

Our affinity measurements for the NS1 PBM interactions with Scribble can be compared to previously measured interactions with other endogenous interactors, such as β-PIX [28,41] β-PIX is an important guanine nucleotide exchange factor for small GTPases, where its membrane-associated localisation is dependent on Scribble interactions, enabling the regulation of cellular processes, such as vesicle trafficking, cytoskeletal organisation, and cell migration [41,42,43,44]. Previous studies have shown that all four of the Scribble PDZ domains can bind to β-PIX through ELISA assays; however, using ITC, we showed that the β-PIX PBM peptide (sequence: PAWDETNL) only bound to SCRIB PDZ1, 2 and 3 [28,41]. The ability of NS1 ESEV to bind to SCRIB PDZ4 may be a significant aspect that allows the influenza A virus NS1 to outcompete endogenous proteins that are unable to engage SCRIB PDZ4. SCRIB PDZ1 displays a higher affinity to β-PIX (3.3 ± 0.3 µM), compared to both NS1 ESEV (21.4 ± 1.7 µM) and ESKV (30.0 ± 0.9 µM) PBMs. In contrast, SCRIB PDZ2 bound tighter to NS1 ESEV (18.7 ± 2.4 µM) and ESKV (9.5 ± 0.7 µM) compared to β-PIX (67.8 ± 7.9 µM). Interestingly, there were no significant differences in the interaction of SCRIB PDZ3 (β-PIX:14.7 ± 2.1 µM, ESEV: 13.1 ± 1.6 µM, ESKV:12.1 ± 0.9 µM). Our affinity measurements indicate that β-PIX has a preference for SCRIB PDZ1, whereas NS1 ESEV and ESKV lean towards SCRIB PDZ4 and PDZ2, respectively, thus providing a mechanism where NS1 can selectively outcompete the endogenous β-PIX for specific Scribble PDZ domains.

When complexes of Scribble PDZ domains are superimposed, no overt differences in the mode of binding between the endogenous cellular and the viral interactor are obvious. SCRIB PDZ1:ESEV and SCRIB PDZ1:ESKV superimposed with SCRIB PDZ1:β-PIX with RMSD values of 0.77 and 0.75 Å, respectively. SCRIB PDZ1:ESKV Chain A superimposed with SCRIB PDZ3:β-PIX showed an RMSD of 0.57 Å. Consequently, all differences between these complexes are due to specific interactions during engagement with the PDZ domains. The −1 position of β-PIX N645 forms a hydrogen bond with S741^PDZ1^, whereas the NS1 PBM does not interact with SCRIB PDZ1 from the −1 position; instead, the −2 position S223 appears to be the important position, forming a hydrogen bond with I742^PDZ1^. Additionally, S1026^PDZ3^ interacts with the β-PIX W641, whereas S1026^PDZ3^ is not involved in the SCRIB PDZ3:ESKV complex; instead, a K1040^PDZ3^-E222^ESKV^ hydrogen bond is formed.

In summary, we show that SCRIB PDZ domains display differential binding affinities for the influenza A virus NS1 and that subtle differences in the detailed interactions drive the differences in affinity observed for PDZ1–4. Our findings provide a structural basis for avian influenza A virus NS1 subversion of SCRIB-mediated polarity signalling, and will form the platform for future structure-guided investigations to understand how the differential ability of individual SCRIB PDZ domains to engage NS1 impacts the control of cell polarity during viral replication.

## Figures and Tables

**Figure 1 viruses-14-00583-f001:**
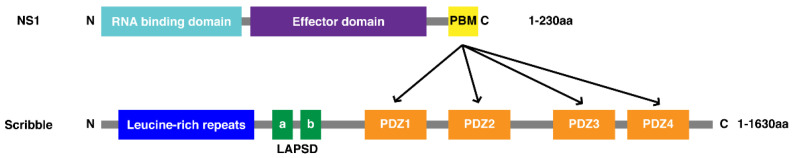
Domain organisation of full-length NS1 and interactions pattern for NS1 PBM motif with full-length SCRIB.

**Figure 2 viruses-14-00583-f002:**
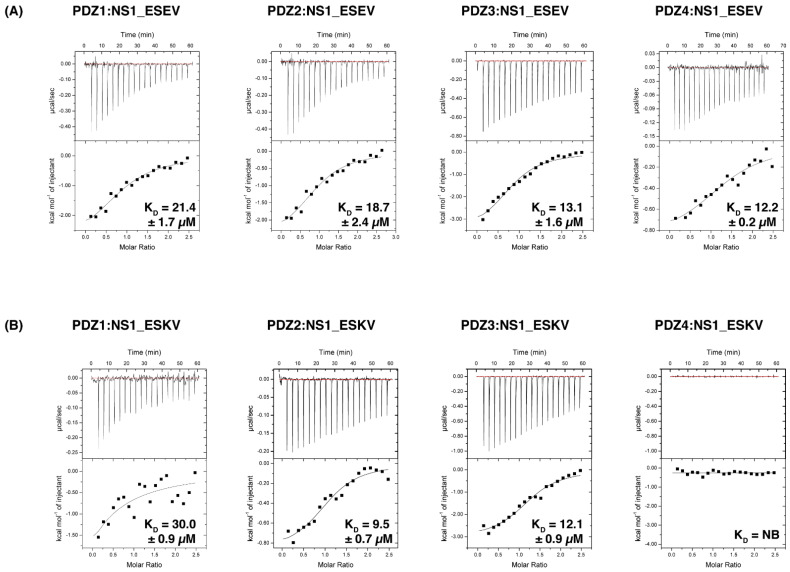
Interactions of the Scribble PDZ domains with NS1 PBM peptides. (**A**) SCRIB individual PDZ domains with NS1 ESEV PBM peptides. (**B**) SCRIB individual PDZ domains with NS1 ESKV PBM peptides. Affinities were measured using isothermal titration calorimetry (ITC) and the raw thermograms are shown. K_D_ values (in µM) are the means of 3 experiments ± SD.

**Figure 3 viruses-14-00583-f003:**
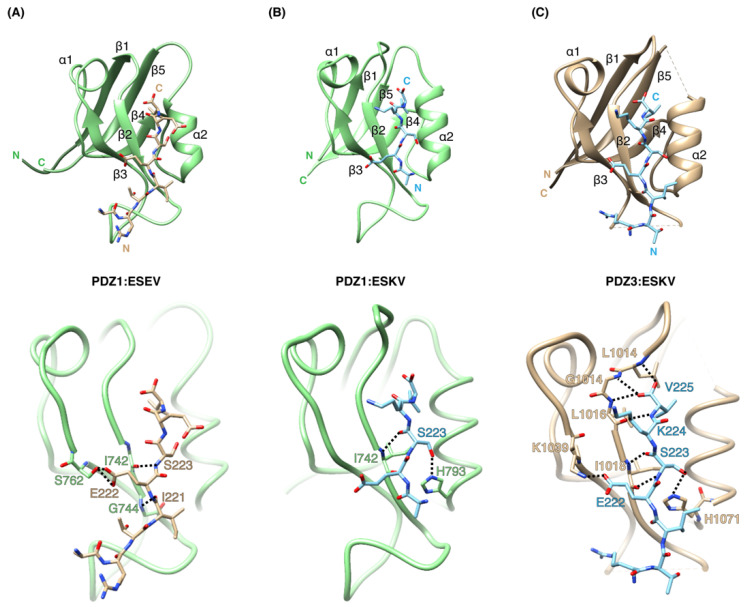
The crystal structures of SCRIB PDZ domains bound to NS1 PBM motif peptides. (**A**) SCRIB PDZ1:ESEV complex with PDZ1 domain (green) shown as a cartoon, and NS1 ESEV peptide (gold) in the top panel. Detailed interactions are shown in the bottom panel. (**B**) SCRIB PDZ1:ESKV complex, with PDZ1 domain (green) shown as a cartoon, and NS1 ESKV peptide (blue) as sticks in the top panel. Detailed interactions are shown in the bottom panel. (**C**) SCRIB PDZ3:ESKV complex with SCRIB PDZ3 domain (gold) shown as a cartoon, and NS1 ESKV peptide (blue) as sticks in the top panel. Detailed interactions are shown in the bottom panel. Interactions are denoted as black dotted lines.

**Figure 4 viruses-14-00583-f004:**
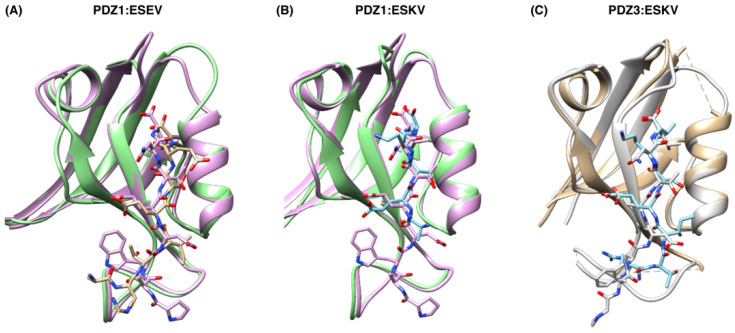
Superimposition of SCRIB PDZ1:NS1 complexes with previously determined SCRIB PDZ structures in complex with host cell β-pix PBM peptides, represented as ribbons and sticks. (**A**) PDZ1:ESEV (green:yellow) superimposed with SCRIB PDZ1:β-pix (pink, PDB ID 5VWK), RMSD is 0.77 Å. (**B**) SCRIB PDZ1:ESKV (green:light blue) superimposed with SCRIB PDZ3:β-PIX (pink, PDB ID 5VW1) and SCRIB PDZ3apo (forest green) RMSD is 0.75 Å. (**C**) PDZ1:ESKV Chain A (gold:light blue) superimposed with SCRIB PDZ3:β-PIX (light grey, PDB ID 5VW1), RMSD is 0.57 Å.

**Figure 5 viruses-14-00583-f005:**
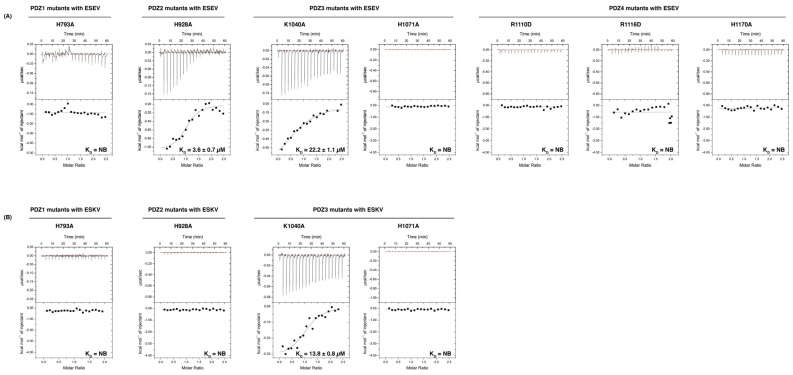
Binding profiles of mutant SCRIB PDZ domain interactions with NS1 PBM peptides. Isolated mutant SCRIB PDZ domain interactions with (**A**) NS1 ESEV PBM peptides or (**B**) NS1 ESKV PBM peptides. Each profile is represented by a raw thermogram, and a binding isotherm fitted with a one-site binding model (bottom panels). K_D_: dissociation constant (in µM); ±: standard deviation (SD); NB: no binding. Each of the values was calculated from at least three independent experiments.

**Figure 6 viruses-14-00583-f006:**
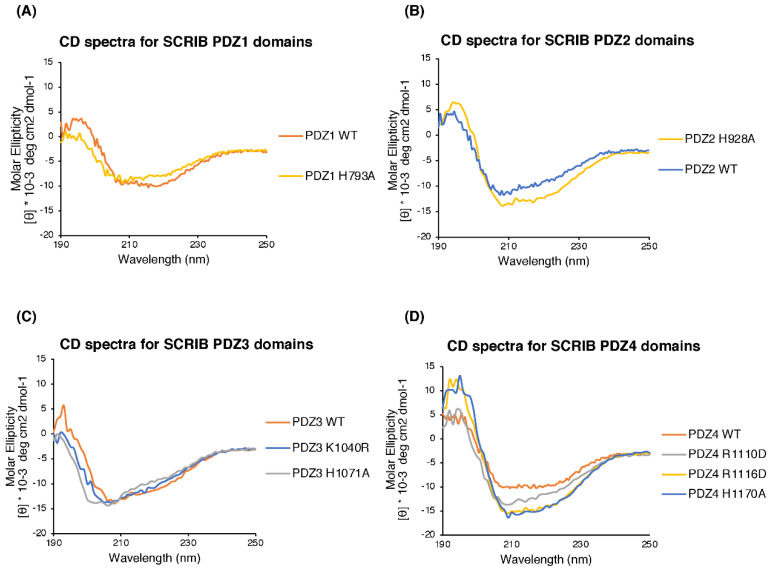
Circular dichroism spectroscopy of wild-type and mutant SCRIB PDZ domains. **(A)** Spectra for Scrib PDZ1, (**B)** spectra for Scrib PDZ2, (**B)** spectra for Scrib PDZ3 and (**D)** spectra for Scrib PDZ4. Circular dichroism spectra recorded for wild-type and mutant SCRIB PDZ1, 2, 3 and 4 domains indicated that there were no major spectral differences between the proteins, with both wild-type and mutant spectra showing the characteristic features indicative of a fold containing mixed alpha/beta secondary structure elements, suggesting that they were similarly folded, with mutations not leading to unfolding of the PDZ domains.

**Figure 7 viruses-14-00583-f007:**
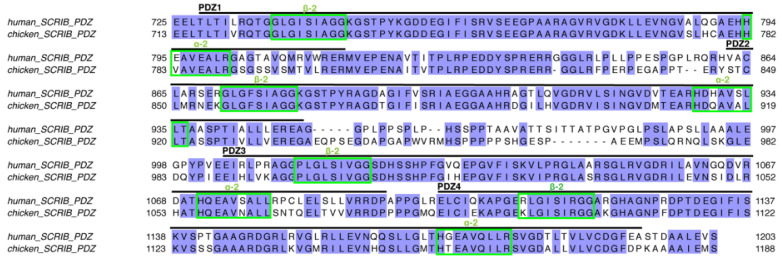
Sequence alignment of human and avian Scribble (Uniprot accession number Q14160 for human SCRIB and Genbank accession number XM_040696172.1 for chicken SCRIB). Conserved residues are shaded in lilac, key regions mediating interactions with NS1 are boxed in green.

**Table 1 viruses-14-00583-t001:** Overview of NS1 PBM motif peptides. (A) Sequence alignment of NS1 peptides and superpeptide created with Clustal Omega 1.2.4 multiple sequence alignment (https://www.ebi.ac.uk/Tools/msa/clustalo/# accessed on 9 August 2018). An * (asterisk) indicates positions which have a single, fully conserved residue; . (period) indicates conservation between groups of weakly similar properties and : (colon) means that conserved substitutions are observed. Residues coloured in red are physiochemically small and hydrophobic (i.e., AVFPMILW). Blue is acidic (i.e., DE), Magenta is basic (i.e., RK). Green is an amine (i.e., STYHCNGQ).

NS1 Peptide Origin	Peptide Sequence									
Vietnam_H5N1	K	M	A	R	T	I	E	S	E	V
Hubei_H5N1	K	M	A	R	T	I	E	S	K	V
Brevig_H1N1	K	M	A	R	T	I	K	S	E	V
Memphis_H3N2	K	M	A	R	T	A	R	S	K	V
Puerto Rico_H1N1	E	M	A	G	T	I	R	S	E	V
HongKong_H5N1	K	M	E	R	T	I	E	P	E	V
	:	*			*		.		:	*

**Table 2 viruses-14-00583-t002:** Interactions of SCRIB PDZ domains with influenza A virus_NS1 peptides. All affinities were measured using isothermal titration calorimetry, with K_D_ values given in µM as a mean of three independent experiments with SD. NB stands for ‘no binding’ and N is the average stoichiometry. NB = no binding. ^$^ = negative control. Mutants were not measured for PBM peptides that did not bind to WT SCRIB PDZ domains.

	PBM Peptides											
SCRIB PDZ Constructs	Vietnam (ESEV)		Hubei (ESKV)		Brevig (KSEV)		Memphis (RSKV)		Puerto Rico (RSEV)		Hong Kong ^$^ (EPEV)	
	N	KD (µM)	N	KD (µM)	N	KD (µM)	N	KD (µM)	N	KD (µM)	N	KD (µM)
PDZ1	1.1 ± 0.1	21.4 ± 1.7	1.2	30.0 ± 0.9	NB	NB	NB	NB	NB	NB	NB	NB
PDZ2	1.0 ± 0.03	18.7 ± 2.4	1.1	9.5 ± 0.7	NB	NB	NB	NB	NB	NB	NB	NB
PDZ3	1.1 ± 0.1	13.1 ± 1.6	1.1	12.1 ± 0.9	NB	NB	NB	NB	NB	NB	NB	NB
PDZ4	1.0 ± 0.1	12.2 ± 0.2	NB	NB	NB	NB	NB	NB	NB	NB	NB	NB
PDZ1 H793A	NB	NB	NB	NB								
PDZ2 H928A	1.1 ± 0.1	3.6 ± 0.7	NB	NB								
PDZ3 K1040A	1.1 ± 0.1	22.2 ± 1.1	1.1	13.8 ± 0.8								
PDZ3 H1071A	NB	NB	NB	NB								
PDZ4 R1110D	NB	NB										
PDZ4 R1116D	NB	NB										
PDZ4 H1170A	NB	NB										

**Table 3 viruses-14-00583-t003:** X-ray crystallographic data collection and refinement statistics. Values in parentheses are for the highest resolution shell.

Data Collection	SCRIB PDZ1: ESEV	SCRIB PDZ1: ESKV	SCRIB PDZ3: ESKV
Space group	I2	C2	C2
No. of molecules in AU	2	1	4
Cell dimensions			
a, b, c (Å)	56.91; 57.01; 61.60	58.54; 51.20; 27.87	77.84; 77.72; 64.75
α, β, γ (°)	90.00; 117.372; 90.00	90.00; 90.55; 90.00	90.00; 94.16; 90.00
Wavelength (Å)	0.95	0.95	0.95
Resolution (Å)	39.74 –3.50 (3.63–3.50	29.26–2.06 (2.13–2.06)	42.94–2.83 (2.93–2.83)
R_sym_ or R_merge_	0.08177 (0.1554)	0.0176 (0.068)	0.05437 (0.5045)
I/σI	6.63 (4.20)	13.58 (6.80)	9.79 (1.94)
CC (1/2)	0.982 (0.94)	0.999 (0.987)	0.999 (0.866)
Completeness (%)	97.52 (99.10)	99.90 (100.0)	96.63 (94.74)
Multiplicity	1.9 (1.9)	2.0 (2.0)	1.9 (1.8)
Wilson B-factor	43.63	30.93	56.7
Refinement			
Resolution (Å)	30.47–3.50 (3.63–3.50)	29.26–2.06 (2.13–2.06)	42.94–2.84 (2.94–2.84)
No. reflections	2199 (220)	10215 (1022)	16752 (1584)
R_work_/R_free_	0.242/0.266	0.277/0.279	0.289/0.295
No. atoms			
Protein	1423	689	2620
Ligand/ion	0	0	0
Water	0	43	0
B-factors			
Protein	26.32	48.46	48.84
Ligand/ion	0	0	0
Water	0	42.46	0
R.m.s. deviations			
Bond lengths (Å)	0.002	0.006	0.003
Bond angles (°)	0.55	0.9	0.59

## Data Availability

Data supporting the findings of this manuscript are available from the corresponding authors upon reasonable request. Coordinate files were deposited at the Protein Data Bank (https://www.rcsb.org/) (accessed on 1 July 2021) using accession codes 7QTO, 7QTP and 7QTU for SCRIB PDZ1:ESEV, SCRIB PDZ1:ESKV and SCRIB PDZ3:ESKV, respectively. The raw X-ray diffraction data were deposited at the SBGrid Data Bank [45] (https://data.sbgrid.org/data/ accessed on 20 January 2022) using their PDB accession codes 7QTO, 7QTP and 7QTU for SCRIB PDZ1:ESEV, SCRIB PDZ1:ESKV and SCRIB PDZ3:ESKV, respectively.

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
