# Peer review of "Structural Basis of the Avian Influenza NS1 Protein Interactions with the Cell Polarity Regulator Scribble"

_viruses, 2022, doi:10.3390/v14030583_

Round 1
Reviewer 1 Report
In this manuscript by Airah Javorsky et al, the authors examined the interaction of the PDZ-binding motif (PBM) of influenza A virus NS1 protein with the four PDZ motifs of the protein Scribble. Indeed, NS1 – or at least some sequence variants of the viral protein- was shown to interact with Scribble, a ~1600-residue protein that is involved in the maintenance of cell polarity.
Main findings
- Each of the four PDZ domains of Scribble was bacterially expressed then purified, along with a set of variants harboring substitutions of critical residues
- Six 10-mer peptides were assayed for their interactions, in vitro, with the different PDZ domains
- The amino-acid sequence of the six 10-mer peptides correspond to distinct variations of the C-teminus of NS1, that are characteristic of either avian H5N1 virus, human H1N1 or human H3N2 viruses. The four C-terminal residues make up the PDZ-binding motif, and it had been shown that variations in the sequence of these four amino-acids dramatically impacts the interaction of NS1 with Scribble.
- The interactions were monitored and characterized in vitro using isothermal titration calorimetry (ITC)
- The crystal structure of some PBM-PDZ complexes were resolved through X-ray diffraction.
The manuscript is well written and scientifically sound. It provides some interesting information. However the interpretation of the data is somewhat excessive, and the authors should emphasize the fact that it is based exclusively on in vitro data that were gathered in experimental conditions that do not represent the biological conditions.
The title is a little bit excessive, as if NS1’s PBM could by itself mediate the subversion of cell polarity. There are no in vivo data to support that claim, and in fact the manuscript is essentially a structural analysis, purely in vitro, of the interaction with PDZ-domains of six distinct 10-mer peptides derived from NS1’s C-terminus. Further, the interactions were measured in the presence of very high concentrations of the two partners (micromolar concentrations of 900 and 75 for the peptides and the recombinant PDZ domains, respectively). These very high concentrations are several orders of magnitude above the concentrations of the protein partners in biological conditions.
Major remarks
The authors put too much emphasis on the role of the PDZ-binding motif of NS1, as if the whole pathogenicity of influenza viruses could be ascribed to this PBM (notably lines 107-111). Further, they do not even mention the fact that there are some variations in the position of NS1’s stop-codon, resulting in a large variety of NS1’s C-terminus: some NS1 variants harbor a C-terminal extension (their length is 237 residues), while others are truncated ~10 to 20 residues upstream of the would-be PBM motif.
Lines 68-69. Even when considering Figures 3 and 4, it is difficult to understand the description of the fold “two alpha-helices wrapped around beta-strands”
Lines 93-95. Inappropriate sentence “Several major PBM sequences have been identified…”. This sentence is somewhat contradictory with the data: several C-terminal sequences do not bind the PDZ domains of Scribble, and even the references (22, 24, 25) provide no support as to the PBM quality of the sequences cited in the sentence.
Lines 99-101 : “an NS1 protein with the ESEV PBM characterized by a single amino-acid change at position 92”. This is nonsense: position 92 does not belong to the C-terminal PBM motif of NS1, and nowhere do the authors (Seo et al., ref 26) cite the word PBM or PDZ. Further, the data in that article (Seo et al. 2002) were challenged by other articles, and in 2012 Nature Medicine published an Erratum (Nat Med. 18(10): 1592), alerting readers about facts that affect the validity of its conclusion
Lines 232-33: in the cited reference [22], the authors did not investigate the binding of a peptide with SCRIB. Instead their assay used a GST-fused, full-length NS1.Further, their assay was only qualitative (it binds or it doesn’t bind). Therefore, although ref 22 is informative, one should be cautious in comparing the present data with those of ref 22.
Figures 2 and 5. The authors should try to interpret the shape of the binding curves. Further, one would expect a plateau, which presumably is required to calculate the dissociation constant. Some curves seem not to reach this plateau.
Table 2 and lines 167-168. How was the stoichiometry established? Further, is the “one site binding model” appropriate when the stoichiometry exceeds 1?
Lines 272-280 and Figure 3 (and lines 357, 404, 406): numbering of amino-acids should be relative to either Udorn or Brevig, not relative to the H5N1 NS1s, since these harbor a deletion of the five residues 80-84. Alternatively, numbering could be relative to the PBM itself. In brief, one should write either E227SEV230, or E-3SEV0, but not E222SEV225
Fig 4. The peptides in the three structures are presumably the same as those in Figures 3,ABC. However they seem to be longer in Figure 4 (additional residues seem to be present). Please explain or comment the difference (in the legend to Figure 4), or remove the additional residues. Further, do these additional amino-acids have no impact on the interaction?
Figure 6. CD curves are somewhat difficult to interpret by a non specialist. The authors should try to explain what exactly allows them to conclude that “there were no major spectral differences between the proteins” (line 313).
Lines 316-17. The first sentence of the discussion implies that NS1 was indeed shown to influence SCRIB signaling. If this is indeed the case, the authors should provide at least one reference supporting that fact.
Line 322. “Unexpectedly” is excessive, since, as stated before and as the authors recognize, one must be cautious when doing this comparison (and indeed ref 32 shows that other residues in full-length NS1 can impact its interactions with PDZ domains).
Minor remarks
Lines 22-23: it would be more informative to write “…using isothermal titration calorimetry (ITC) that ESKV and KSEV bind to three or four of the four PDZ domains of SCRIB”
Line 57 …sixteen LRRs (leucine-rich repeats)..
Line 114;;; “The cDNAs encoding the human Scribble…”
Table 2, second line : the name of the strain (RSEV) is Udorn, not Urdon.
Line 333. Substitution (not substation)
Line 350. The majority of the of our ??
Lines 398-400. Verify grammar (“…as an RMSD is 0.77…showed an RMSD is 0.57”
Author Response
Response to reviewers:
Reviewer #1:
“The title is a little bit excessive, as if NS1’s PBM could by itself mediate the subversion of cell polarity.”
Response: We have amended the title to focus on the fact that we define the structural basis of NS1 interactions with Scribble. We also preface the discussion with a statement to indicate that our measurements were performed in vitro: “To understand the ability of the Influenza A virus protein NS1 to influence SCRIB signalling as an exogenous interaction partner of SCRIB, we systematically examined the interactions of individual SCRIB PDZ domains with the PBM peptides of several Influenza A virus strains in vitro using ITC.”
The authors put too much emphasis on the role of the PDZ-binding motif of NS1, as if the whole pathogenicity of influenza viruses could be ascribed to this PBM (notably lines 107-111). Further, they do not even mention the fact that there are some variations in the position of NS1’s stop-codon, resulting in a large variety of NS1’s C-terminus: some NS1 variants harbor a C-terminal extension (their length is 237 residues), while others are truncated ~10 to 20 residues upstream of the would-be PBM motif.
Response: This is not correct. We do not state anywhere in the manuscript that the whole pathogenicity of influenza viruses is solely due to PDZ binding motif interactions. As stated in line 103 in the manuscript “These sequences all appear to contribute to the pathogenicity…”. As I am sure the reviewer is aware viruses are complex organisms featuring a multitude of biomolecules, and it is ultimately the totality of these biomolecules that are responsible for overall pathogenicity. However, we have added an additional statement that discusses existing variations in Influenza A virus NS1 protein C-termini and clarifies differences between avian and human NS1 proteins as follows: “Whilst the four C-terminal residues of NS1 were shown to make species-specific contributions to Influenza A virus virulence [24], sequence variation has been seen at the NS1 C-terminus including premature termination, yielding NS1 proteins that lack PBM sequences. For avian Influenza A NS1, ~90% of isolates contain an ESKV or ESEV PBM motif at the C-terminus [25], whereas in human Influenza A the RSKV PBM motif is more widely found [26].”.
Lines 68-69. Even when considering Figures 3 and 4, it is difficult to understand the description of the fold “two alpha-helices wrapped around beta-strands”
Response: We have clarified the description of the PDZ domain as follows: “PDZ domains are usually approximately 90 amino acids in size and consist of 5-6 β-strands and 2-3 α-helices with a conserved fold comprising a canonical ligand binding groove formed by the α2 helix and the β2, β4 and β5 strands. This binding groove allows PDZ domains to recognize short amino acid motifs at the extreme C-termini of ligand proteins coined PDZ binding motifs (PBMs), however certain PDZ domains have also been shown to recognise internal sequence motifs [18,19].”
Lines 93-95. Inappropriate sentence “Several major PBM sequences have been identified…”. This sentence is somewhat contradictory with the data: several C-terminal sequences do not bind the PDZ domains of Scribble, and even the references (22, 24, 25) provide no support as to the PBM quality of the sequences cited in the sentence.
Response: We have amended the sentence as follows: “Human influenza A virus protein NS1 has been shown to harbor multiple C-terminal sequences that conform to the consensus sequence of class I PDZ binding motifs: RSKV (seen in H3N2), ESKV (found in H9N2, H5N1, H5N2), KSEV (associated with H1N1), ESEV (from H5N1) and RSEV (from H1N1) [22,24,25].”
We note that ref 22 (Liu et al JVI 2010) specifically examines the interaction of the ESEV PBM motif with Scribble and discusses RSKV is from H3N2 A/Memphis/14/1998, whilst ref 23 (Hale et al J Gen Virol) discusses the appearance of a C-terminal extension to NS1 without specifically referring to it as a PDZ binding motif. We have removed references 24 and 25.
Lines 99-101 : “an NS1 protein with the ESEV PBM characterized by a single amino-acid change at position 92”. This is nonsense: position 92 does not belong to the C-terminal PBM motif of NS1, and nowhere do the authors (Seo et al., ref 26) cite the word PBM or PDZ. Further, the data in that article (Seo et al. 2002) were challenged by other articles, and in 2012 Nature Medicine published an Erratum (Nat Med. 18(10): 1592), alerting readers about facts that affect the validity of its conclusion
Response: We thank the reviewer for alerting us to the challenge to the data we refer to. In light of this we have removed the entire statement.
Lines 232-33: in the cited reference [22], the authors did not investigate the binding of a peptide with SCRIB. Instead their assay used a GST-fused, full-length NS1.Further, their assay was only qualitative (it binds or it doesn’t bind). Therefore, although ref 22 is informative, one should be cautious in comparing the present data with those of ref 22.
Response: We have changed the sentence for improved clarity as follows: “Interestingly, these interactions contrast with previously observed interactions from GST pull-down studies that suggested the ESEV PBM sequence in the context of recombinant full-length NS1 was only able to bind to a tandem of PDZ1 and PDZ2 [22].”
Figures 2 and 5. The authors should try to interpret the shape of the binding curves. Further, one would expect a plateau, which presumably is required to calculate the dissociation constant. Some curves seem not to reach this plateau.
Response: There are no unusual features in the ITC thermograms that require additional interpretation. Titration curves of interactions in the mid micromolar range do not produce plateaus as has been shown by many others (for example see Mao et al Cell Discovery 2022 PMID 35075146).
Table 2 and lines 167-168. How was the stoichiometry established? Further, is the “one site binding model” appropriate when the stoichiometry exceeds 1?
Response: Stoichiometry is affected by protein and peptide concentration measurements. We have added SD values for the stoichiometry to reflect the fact that the values are obtained from 3 independent measurements. One site binding model is entirely appropriate, as is seen in our crystal structures the interaction is a 1:1 interaction.
Lines 272-280 and Figure 3 (and lines 357, 404, 406): numbering of amino-acids should be relative to either Udorn or Brevig, not relative to the H5N1 NS1s, since these harbor a deletion of the five residues 80-84. Alternatively, numbering could be relative to the PBM itself. In brief, one should write either E227SEV230, or E-3SEV0, but not E222SEV225
Response: We agree that we may be confusing the reader, and have now amended the relevant numbering as follows: “We determined crystal structures of SCRIB PDZ1 bound to NS1 ESEV PBM (A281‐R219-T220‐I221‐E222‐S223‐E224‐V225; Fig 3), SCRIB PDZ1 bound to NS1 ESKV PBM (A281‐R219-T220‐I221‐E222‐S223‐K224‐V225; Fig 3) and PDZ3 bound to NS1 ESKV PBM (A281‐R219-T220‐I221‐E222‐S223‐K224‐V225; Fig 3). “
We have amended the manuscript to reflect the new numbering convention.
Fig 4. The peptides in the three structures are presumably the same as those in Figures 3,ABC. However they seem to be longer in Figure 4 (additional residues seem to be present). Please explain or comment the difference (in the legend to Figure 4), or remove the additional residues. Further, do these additional amino-acids have no impact on the interaction?
Response: We show the same number of residues in the structures of our Scribble:NS1 PBM complexes in Figure 3 and Figure 4. The apparent discrepancy is due to the fact that we superimposed the Scribble PDZ1:b-pix complex onto our Scribble:NS1 PMB complexes in Figure 4 to illustrate the fact that there are no overt structural differences between these complexes. The “extra” residues the reviewer is referring to are from b-pix, not from NS1.
Figure 6. CD curves are somewhat difficult to interpret by a non specialist. The authors should try to explain what exactly allows them to conclude that “there were no major spectral differences between the proteins” (line 313).
Response: We have amended the legend for Figure 6 referring to the CD data as follows: “Circular dichroism spectra recorded for wild type and mutant SCRIB PDZ1, 2, 3 and 4 domains indicated that there were no major spectral differences between the proteins, with both wild-type and mutant spectra showing the characteristic features indicative for a fold containing mixed alpha/beta secondary structure elements, suggesting that they were similarly folded, with mutations not leading to unfolding of the PDZ domains.
Lines 316-17. The first sentence of the discussion implies that NS1 was indeed shown to influence SCRIB signaling. If this is indeed the case, the authors should provide at least one reference supporting that fact.
Response: We now cite Golebiewski et al JVI 2011 to support this statement, who showed that the interaction of ESEV PBM with Dlg and Scribble disrupts tight junctions.
Line 322. “Unexpectedly” is excessive, since, as stated before and as the authors recognize, one must be cautious when doing this comparison (and indeed ref 32 shows that other residues in full-length NS1 can impact its interactions with PDZ domains).
Response: We removed “unexpectedly” from the sentence.
Minor remarks
Lines 22-23: it would be more informative to write “…using isothermal titration calorimetry (ITC) that ESKV and KSEV bind to three or four of the four PDZ domains of SCRIB”
Response: We amended this section in the abstract as follows: “We now show using isothermal titration calorimetry (ITC) that ESKV binds to SCRIB PDZ domains 1,2 and 3 and that ESEV unexpectedly displayed an affinity towards all four PDZ domains.”
Line 57 …sixteen LRRs (leucine-rich repeats)..
Response: We corrected this.
Line 114;;; “The cDNAs encoding the human Scribble…”
Response: We amended the sentence to “The synthetic codon optimized cDNAs encoding for the human Scribble…”
Table 2, second line : the name of the strain (RSEV) is Udorn, not Urdon.
Response: We changed the strain to Puerto Rico. Inadvertently the column was labelled Udorn, but as (correctly) described in the methods section the sequence is associated with the A/Puerto Rico/8/1943/H1N1 strain.
Line 333. Substitution (not substation)
Response: We corrected this.
Line 350. The majority of the of our ??
Response: We corrected this.
Lines 398-400. Verify grammar (“…as an RMSD is 0.77…showed an RMSD is 0.57”
Response: We corrected this.
Reviewer 2 Report
The paper by Airah Javorsky and co-authors defined the structural basis for the interactions of SCRIB PDZ1 domain with ESEV and ESKV PBM motifs as well as SCRIB PDZ3 with the ESKV PBM motif. Although the study provided a structural basis for Influenza A NS1 subversion of SCRIB mediated polarity signaling and will form the platform for future structure-guided investigations to understand how the differential ability of individual SCRIB PDZ domains to engage NS1 impacts the control of cell polarity during viral replication, there are also some deficiencies in this paper as below.
1、In line 185th:"belonging to" should be "belonged to".
2、In Figure 2, can you explain why others data all fit to the curve line except "PDZ1:NS1_ESKV".
3、In Figure 3, the illustration is a little confused, there are some improper words, such as "below".
4、In Figure 6, the arrangement of figure legends should be consistent, the Y axis calibration should not be diverse, some information lost in Figure 6.
Author Response
In line 185th:"belonging to" should be "belonged to".
Response: We corrected this.
In Figure 2, can you explain why others data all fit to the curve line except "PDZ1:NS1_ESKV".
Response: This is due to two factors: the data range allowed for plotting the curve and that we had more variability in this run compared to other interactions measured. We have replaced the thermogram with one obtained from a different run.
In Figure 3, the illustration is a little confused, there are some improper words, such as "below".
Response: We amended the Figure legend as follows: “SCRIB PDZ1:ESEVcomplex with PDZ1 domain (green) shown as a cartoon and NS1 ESEV peptide (gold) in the top panel. Detailed interactions are shown in the bottom panel.”
In Figure 6, the arrangement of figure legends should be consistent, the Y axis calibration should not be diverse, some information lost in Figure 6.
Response: We have amended the figure to show uniform Y scales across all spectra.
Reviewer 3 Report
In the present study, Airah Javorsky and colleagues scrutinised the physical interactions of various strain-specific C-terminal PDZ-binding motifs of influenza A virus (IAV) non-structural protein 1 (NS1) with the four different PDZ domains of human Scribble (SCRIB), a central regulator of cell polarity. Using isothermal titration calorimetry (ITC), the authors evidenced binding of the PDZ-binding motifs (PBM) ESEV and ESKV to at least three SCRIB PDZ domains (PDZ1-3), with the ESEV motif displaying affinity to all four PDZ domains. Using an X-ray diffraction/crystallography approach, they then defined the structural basis for the interactions of some selected PDZ-PDM complexes (PDZ1-ESEV/ESKV, PDZ3-ESKV).
The findings are interesting and contribute to a better understanding of the intricate interplay of NS1 with host cellular proteins.
The manuscript is well written and the methodology is sound. The data are solid and superbly presented.
Major criticism:
The biological relevance of the findings is not sufficiently discussed. The H5N1 PBM tested in this study relate to avian influenza virus/IAV strains for which only accidental human infections are reported. These strains have evolved in and are adapted to avian hosts. The homology between human and avian Scribble proteins/PDZ domains should be analysed, and the relevance of the present findings for IAV infection in the natural animal host should be discussed.
Minor points:
Abstract (lines 22-24): Correct/rephrase the statement (to my understanding KSEV did not show any binding affinity).
IVA is a rather unconventional acronym for influenza A virus. Please refer to IVA.
Please refer to influenza A virus NS1 instead of influenza A NS1.
Introduction (lines 93-95): Correct "Several major PBM sequences have been identified in human influenza A NS1:" to "Several major PBM sequences have been identified in influenza A virus NS1:" (H9N2, H5N1 and H5N2 are avian virus subtypes).
Author Response
The manuscript is well written and the methodology is sound. The data are solid and superbly presented.
Response: We are pleased the reviewer liked our data.
The biological relevance of the findings is not sufficiently discussed. The H5N1 PBM tested in this study relate to avian influenza virus/IAV strains for which only accidental human infections are reported. These strains have evolved in and are adapted to avian hosts. The homology between human and avian Scribble proteins/PDZ domains should be analysed, and the relevance of the present findings for IAV infection in the natural animal host should be discussed.
Response: This is an excellent point. We have generated a sequence alignment comparing human and avian Scrib PDZ domains, and added the following paragraph to the discussion: “Considering that H5N1 originated in birds, we examined the protein sequence conservation of interacting human and avian Scribble PDZ domains (Figure 7). Examination of the main NS1 interaction mediating secondary structure elements from Scrib PDZ domains reveals that all residues involved in hydrogen bonds are conserved, as well as all main residues involved in van der Waals contacts. This suggests that avian Influenza A virus NS1 may be as effective at manipulating Scribble mediated polarity signalling in avian species as in humans.”
Abstract (lines 22-24): Correct/rephrase the statement (to my understanding KSEV did not show any binding affinity).
Response: We amended this section in the abstract as follows: “We now show using isothermal titration calorimetry (ITC) that ESKV binds to SCRIB PDZ domains 1,2 and 3 and that ESEV unexpectedly displayed an affinity towards all four PDZ domains.”
Please refer to influenza A virus NS1 instead of influenza A NS1.
Response: We have changed this throughout manuscript.
Introduction (lines 93-95): Correct "Several major PBM sequences have been identified in human influenza A NS1:" to "Several major PBM sequences have been identified in influenza A virus NS1:" (H9N2, H5N1 and H5N2 are avian virus subtypes).
Response: We have amended the sentence as follows: “Human influenza A virus protein NS1 has been shown to harbor multiple C-terminal sequences that conform to the consensus sequence of class I PDZ binding motifs: RSKV (seen in H3N2), ESKV (found in H9N2, H5N1, H5N2), KSEV (associated with H1N1), ESEV (from H5N1) and RSEV (from H1N1) [22,24,25].”
Round 2
Reviewer 1 Report
Thank you for answering to each of my remarks
Author Response
We are pleased the reviewer is satisfied with our responses.
Reviewer 2 Report
- In the whole article, you amended "Influenza A" to "Influenza A virus" in some places, while there were still some others not modified.
- In line 144, you used "20hrs" to present incubation time, while in lines 148 and 149 you used "hours"? You should check your full paper seriously.
- In line 121, "range from" is not appropriate enough, how about "cover"?
- In your revision manuscript, the flaws which was pointed out in Figure 6 last time still existed.
Author Response
- In the whole article, you amended "Influenza A" to "Influenza A virus" in some places, while there were still some others not modified.
Response: We have amended all instances of Influenza A to Influenza A virus in the manuscript.
2. In line 144, you used "20hrs" to present incubation time, while in lines 148 and 149 you used "hours"?
Response: We now uniformly use hrs.
3. In line 121, "range from" is not appropriate enough, how about "cover"?
Response: We have amended the sentence as follows: “The severity of the H5N1 influenza A virus can cover diarrhoea, encephalitis, cytokine storm, oedema within the lungs to organ failure and death [27]”
4. In your revision manuscript, the flaws which was pointed out in Figure 6 last time still existed.
Response: We apologize for this, the amended Figure is now included.